# ALIGNING MULTIMODAL MODELS FOR CLINICAL REASONING USING RULE-BASED REWARDS

## ABSTRACT

Vision-Language Models (VLM) can support clinicians by analyzing medical images and engaging in natural language interactions to assist in diagnostic and treatment tasks. However, VLMs often exhibit "hallucinatory" behavior, generating textual outputs not grounded in contextual multimodal information. This challenge is particularly pronounced in the medical domain, where we do not only require VLM outputs to be accurate in single interactions but also to be consistent with clinical reasoning and diagnostic pathways throughout multi-turn conversations. For this purpose, we propose a new alignment algorithm that uses *rule-based representations* of clinical reasoning to ground VLMs in medical knowledge. These representations are utilized to **(i)** generate visual instruction tuning data at scale, simulating clinician-VLM conversations with demonstrations of clinical reasoning, and **(ii)** to derive a rule-based reward function that automatically evaluates the clinical validity of VLM responses throughout clinician-VLM interactions. Our algorithm eliminates the need for human involvement in training data generation or reward model construction, reducing costs compared to standard reinforcement learning with human feedback (RLHF). We apply our alignment algorithm to develop Dr-LLaVA, a conversational VLM finetuned for analyzing bone marrow pathology slides, demonstrating strong performance in single and multi-turn medical conversations.

## 1 INTRODUCTION

Vision-language models (VLMs) (1–3), which integrate large language models (LLMs) (4–8) with vision encoders, have demonstrated strong capabilities in answering complex questions that require both visual and textual reasoning. In the medical domain, VLMs hold great promise—they could serve as helpful assistants for clinicians, researchers, and trainees, providing an interactive natural language interface for the analysis of medical images within clinical workflows (9–14). However, the practical utility of present VLMs is significantly limited by their tendency to "hallucinate". In this context, hallucination refers not only to instances where the model generates responses ungrounded in visual input but also to cases where, in multi-turn interactions, its responses are incoherent, contradictory, or misaligned with diagnostic pathways and domain knowledge.

The currently predominant methods to reduce hallucinations in VLMs such as Reinforcement Learning from Human Feedback (RLHF) (15–18) are not well-suited for the multimodal medical context. Using RLHF to align VLMs with visually-grounded clinical reasoning requires multimodal training data showcasing the reasoning process within multi-turn QA dialogues. These datasets are not readily available in health systems. Synthesizing these datasets and collecting clinician feedback on VLM responses is bottlenecked by the expertise of medical professionals. Unlike the LLaVA-RLHF model in (18), which gathered human feedback from non-expert crowdworkers for simple, common-sense visual QA tasks, this process cannot be scaled without the involvement of clinicians. Due to these limitations, specialized medical VLMs like LLaVA-Med (9) and PathChat (2) have been limited to supervised finetuning, relying on automatically generated QA tasks using image captions. Moreover, both existing general-purpose and medical VLMs have only been finetuned for single-turn QA, rather than for multi-turn conversations that convey complex and interactive clinical reasoning.

In this work, we leverage the key insight that many clinical reasoning pathways can be formalized as a *hierarchical set of rules*. This formalization enables the decomposition of ambiguous medical inquiries into a sequence of logical steps, where the outcomes of earlier sub-analyses constrain the set of permissible diagnoses in subsequent stages. For instance, the example conversation in Fig. 1 follows a typical analysis workflow for bone marrow pathology slides. It starts with an assessment of image quality, then analyzes cell growth, identifies the types of cells displaying growth, and ultimately derives a final diagnosis. A good VLM should not only arrive at a correct diagnosis but also provide clinically consistent answers throughout the multi-turn conversation. For example, if the model labels an image as being of insufficient quality, we do not want it to proceed to derive further diagnoses. Our

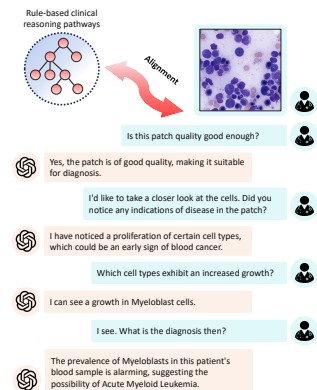

Figure 1: **Rule-based clinical grounding.** Using rule-based models of diagnostic, we align conversational VLMs with valid clinical reasoning.

proposed method leverages these rules to automatically synthesize finetuning datasets of realistic multi-turn VLM-clinician conversations. Furthermore, we design a novel alignment algorithm that extends the RLHF procedure by introducing a reward function that automatically evaluates VLM responses, promoting accurate single-turn responses, while ensuring consistency with correct clinical reasoning across the entire multi-turn dialogue. This enables us to adapt VLMs to multi-turn imaging-based conversational diagnostic tasks, while eliminating the need for human involvement in training data generation or feedback collection. Our method requires only a description of the key clinical reasoning steps necessary to arrive at a specific diagnosis. Such information is for instance readily available from clinical guidelines (19), which are abundant across medical domains (20–24).

We demonstrate the utility of our proposed algorithm by finetuning the LLaVA model (2) to develop Dr-LLaVA, a VLM designed for diagnosing blood cancer using bone marrow pathology images. To this end, we curated a dataset comprising 16,340 bone marrow image patches and generate corresponding multi-turn clinician-VLM conversations. Our results show that Dr-LLaVA outperforms state-of-the-art VLMs in both single- and multi-turn conversational settings. Furthermore, ablation experiments show that our instruction-tuning framework enables Dr-LLaVA to attain high robustness to variations in question sequencing, and to outperform other baselines in identifying and correcting erroneous information in clinician prompts. These findings underscore the value of integrating clinical domain knowledge into fine-tuning approaches using a hybrid rule-based and data-driven method, thereby developing trustworthy and accurate conversational assistants in medicine.

## 2 VISUAL INSTRUCTION TUNING WITH RULE-BASED CLINICAL GROUNDING

Many medical diagnostic processes can be described using a relatively small number of logical rules applied sequentially. Fig. 3(a) presents such a rule-based representation, constructed and adjudicated by an expert pathologist, which outlines each step in the process for diagnosing blood cancer based on bone marrow pathology slides. This decision tree delineates the valid reasoning pathways that VLM responses must adhere to in order to maintain clinical coherence. Formally, we define

$$\mathcal{S} = \big\{(\text{Low image quality}) \rightarrow \text{Inconclusive}, (\text{High image quality} \wedge \text{No abnormality}) \rightarrow \text{Healthy},$$

$$(\text{High image quality} \wedge \text{Abnormality} \wedge \text{Plasma cell proliferation}) \rightarrow \text{Multiple Myeloma}, \dots \big\}, \quad (1)$$

as the subset of all decision rules $\bar{\mathcal{S}}$ that correspond to *valid* reasoning, determined by the representation in Fig. 3(a). Our instruction tuning framework leverages this rule-based representation to (a) synthesize a dataset of clinician-VLM conversations, (b) automatically evaluate clinical consistency of VLM responses, and (c) finetune the VLM to ensure clinical correctness and coherence (Fig. 2).

### 2.1 STEP 1: SYNTHESIZING CLINICIAN-VLM CONVERSATIONS

We synthesize clinician-VLM conversations using a dataset derived from bone marrow aspirate (BMA) whole slide images, annotated by hematopathologists and sourced from the clinical archives

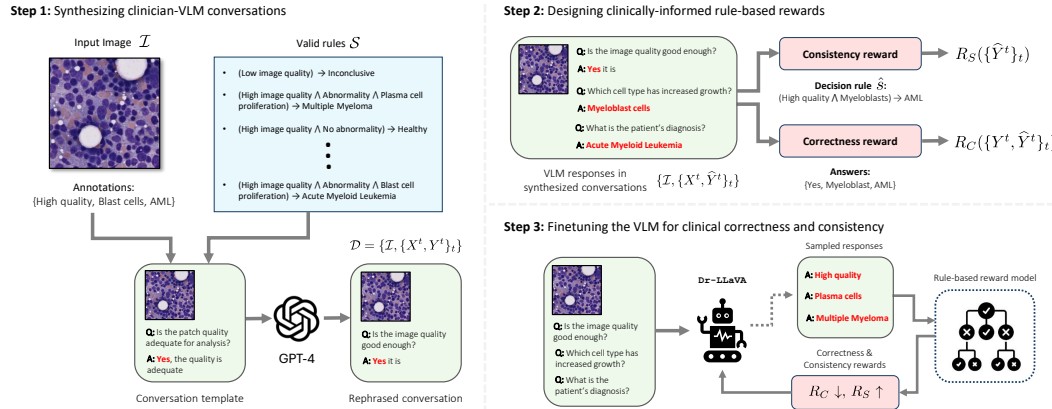

Figure 2: **Pictorial depiction of the alignment algorithm** (a) Multi-turn conversations consistent with rule-based clinical reasoning are generated for each medical image, utilizing GPT-4 for diverse phrasing. (b) A rule-based reward function evaluates VLM responses, checking individual correctness and clinical validity. (c) Using the dataset from (a) and the reward model from (b), a pretrained VLM is finetuned via RL.

of an academic medical center. The dataset includes images indicative of various conditions: blood contamination, particle-enriched contamination, acute myeloid leukemia, multiple myeloma, and healthy states. For each image, we use the hematopathologist's annotations to select the rule from $\mathcal{S}$ that describe the corresponding diagnostic analysis. These rules are then used to construct a multimodal instruction tuning dataset $\mathcal{D} = (\mathcal{I}_i, X_i^t, Y_i^t)_i$, where each image $\mathcal{I}_i$ is paired with multi-turn clinical conversations $X_i^t, Y_i^t$. Each $X_i^t$ represents the $t$-th clinician prompt, and $Y_i^t$ is the corresponding target response, generated by applying textual templates to the image annotations for the respective analysis step (e.g., image quality, cell types, diagnoses). The conversations consist of five question-answer pairs that follow the diagnostic analysis process. To enhance diversity, GPT-4 is employed to generate paraphrased prompts and responses, followed by a rigorous evaluation to ensure no hallucinations are introduced within the template set. An illustration of this dataset synthesis process is provided in Fig. 2. This dataset serves as the basis for our instruction-tuning framework, which combines supervised finetuning and reinforcement learning.

## 2.2 STEP 2: DESIGNING CLINICALLY-INFORMED RULE-BASED REWARDS

In contrast to standard RLHF approaches that rely on human feedback to evaluate ambiguous qualities of model outputs (18; 17; 25), our conversational diagnostic system leverages ruke-based representations (Fig. 3) to convert complex diagnostic questions into a sequence of discrete decisions. This approach enables us to define an efficient keyword-matching algorithm that evaluates VLM responses against specific terms associated with a limited set of admissible answer categories in the decision tree. This facilitates automated evaluation without costly human annotation. A comprehensive list of keywords is provided in Appendix Table B.5.

Given this discrete categorization, we define a reward model that assesses both the correctness of model responses and their alignment with valid clinical reasoning. For an input image $\mathcal{I}_i$ and a sequence of prompted VLM outputs $(X_i^t, \widehat{Y}_i^t)_t$, we compute the reward function as:

$$R((\widehat{Y}_i^t, Y_i^t), \dots, (\widehat{Y}_i^T, Y_i^T)) = \frac{1}{T} \sum_{t=1}^{T} \underbrace{R_C(Y_i^t, \widehat{Y}_i^t)}_{\textbf{Correctness of responses}} + \underbrace{\lambda \cdot R_\mathcal{S}(\{\widehat{Y}_i^t\}_t)}_{\textbf{Consistency with valid reasoning}} + R_l - R_m.$$

(2)

Here, $R_C$ evaluates the accuracy of individual model responses against ground truth, while $R_\mathcal{S}(.)$ assesses whether the VLM's answer sequence aligns with a clinically valid reasoning path. In particular, the rule-based reward function $R_\mathcal{S}(.)$ maps the VLM textual outputs $\{(X_i^t, \widehat{Y}_i^t)\}_t$ to a rule $\hat{s}_i \in \bar{\mathcal{S}}$, and then assigns a reward if the rule is valid, i.e., $\hat{s}_i \in \mathcal{S} \subset \bar{\mathcal{S}}$. The hyperparameter $\lambda$ balances correctness and consistency rewards.

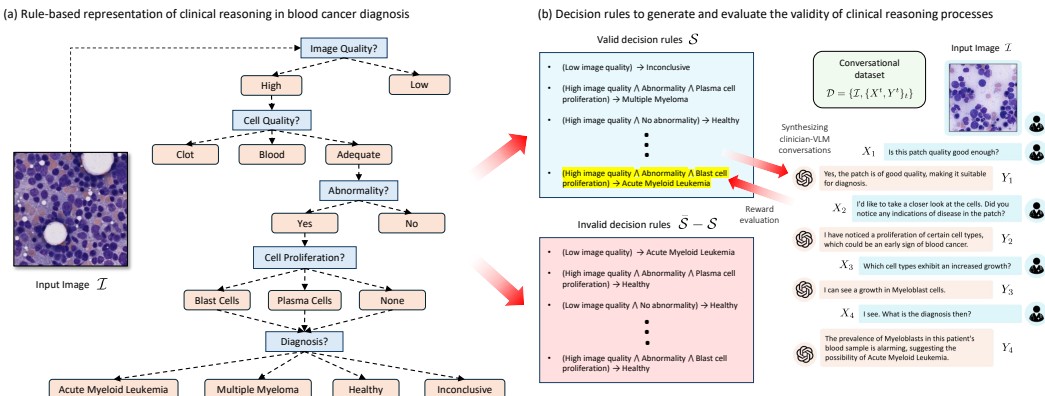

Figure 3: Depiction of our rule-based representation of clinical reasoning in blood cancer diagnosis.

In addition to the correctness and consistency rewards, we use two additional penalties to counteract reward-hacking, similar to (18). First, the penalty term $R_m$ reduces scores for outputs that remain ambiguous and cannot be clearly categorized for a given analysis step. This penalty is quantified by the proportion of ambiguous answers within a conversation. Second, to address the potential for the rule-based rewards to encourage either overly verbose responses that incidentally include relevant keywords, or overly concise, keyword-dense answers, we implement a length-based penalty $R_l$ to discourage significant deviations between the length of the VLM's answer and the target answer length. This approach is designed to promote a conversational style that is both natural and engaging.

## 2.3 STEP 3: FINETUNING THE VLM FOR CLINICAL CORRECTNESS AND CONSISTENCY

We employ a two-stage approach to optimize the VLM for clinical tasks. First, we perform supervised finetuning (SFT) to obtain the initial policy model $\pi_{\text{SFT}}^{\phi}$. To do so, we use the LLaVA architecture (26; 18) and jointly instruction-tune a vision encoder and a pre-trained LLM using token-level supervision to derive a supervised fine-tuned (SFT) model $\pi_{\text{SFT}}^{\phi}$. Following prior work (2; 26), the model is trained based on the LLMs original autoregressive training objective. Specifically, for an answer sequence of length $T$, we compute the probability of the target answer as

$$p(Y_i^t|X_i^t, I_i) = \Pi_{t=1}^T \pi_{\text{SFT}}^{\phi}(y_j|\mathcal{I}_i, \{X_i^{t'}, Y_i^{t'}\}_{t'<t}) \tag{3}$$

where $y_j$ refers to the current prediction token in the answer sequence and $\{X_i^{t'}, Y_i^{t'}\}_{t'<t}$ refers to the tokens in the previous parts of the answer sequence. Throughout this process we keep the vision-encoder fixed and update the weights of the projection layer and the language model to adapt to the clinical domain.

Subsequently, we refine this model using Reinforcement Learning (RL) based on our automatically evaluated rule-based rewards. In the RL stage, we treat $\pi_{\text{SFT}}^{\phi}$ as our initial policy model and train it to generate responses that maximize the reward function $R$, which assesses clinical correctness and consistency. Following (17; 18), we implement Proximal Policy Optimization (PPO) (27) with a per-token Kullback-Leibler (KL) penalty to mitigate reward hacking. This penalty constrains the divergence of the RL-tuned model from that of the SFT model. Given a dataset of medical images, clinical analysis prompts, and their respective answers $\mathcal{D}_{RL} = \{(\mathcal{I}_i, \{X_i^t, Y_i^t\}_t)\}_i$ we define the full finetuning loss as:

$$\mathcal{L}(\pi_{\text{RL}}^{\phi}) = -\mathbb{E}_{(\mathcal{I},X,Y)\in\mathcal{D}_{\text{RL}}, \widehat{Y}\sim\pi_{\text{RL}}(\widehat{Y}|\mathcal{I},X)} \left[ R(\{\widehat{Y}^t, Y^t\}_t) - \beta \cdot D_{\text{KL}}(\pi_{\text{RL}}^{\phi}(\widehat{Y}|\mathcal{I},X)\|\pi_{\text{SFT}}^{\phi}(\widehat{Y}|\mathcal{I},X)) \right]$$

Notably, unlike previous RLHF methods (18), our loss function $\mathcal{L}(\pi_{\text{RL}}^{\phi})$ is computed over the entire multi-turn conversation, as the consistency reward in (2) is evaluated using the full sequence of model responses.

## 3 RELATED WORK

**Vision-Language Models for medicine.** Large Language Models (LLMs) (4; 8; 28–31; 5; 6; 32; 33) have excelled in generating high-quality textual responses across diverse tasks, fueling advancements in chat-based AI assistants (7; 34). Recent work has extended these models to handle multimodal image-text data (35–37), which has led to the emergence of powerful vision-language models (VLM) including OpenFlamingo (38), MiniGPT-4 (39) and LLaVA (9). In the medical domain, the integration of images and texts has been explored in areas such as ultrasound (40; 41), pathology (42; 12), and radiology (43; 44), typically utilizing modality-specific vision encoders. Additionally, recent studies have proposed models that directly finetune state-of-the-art VLMs for medical applications including Med-Alpaca (45), Med-Flamingo (46) and LLaVAMed (9). However, these models solely leverage instruction-tuning with token-level supervision, which can lead to misalignment between image and text modalities, resulting in outputs insufficiently grounded in the visual context (18). Moreover, such approaches do not regularize the model outputs on a conversation-level by incorporating domain knowledge on diagnostic pathways.

**Hallucination in generative models.** In the Natural Language Processing (NLP) literature, "hallucination" is defined as the phenomenon where a model generates content diverging from the original source material (47). With the advent of advanced LLMs, this definition has expanded. As noted in (48), hallucination can manifest in three distinct ways: 1) *Input-conflicting* hallucination, observed in scenarios like machine translation and summarization, where the model's response alters or misinterprets the static context of the user's prompt (49–52); 2) *Context-conflicting* hallucination, where the model's output contradicts its previous responses (53; 54); and 3) *Fact-conflicting* hallucination, in which the generated content conflicts with established factual knowledge (55; 56). Our finetuning framework represents a novel approach to address context-conflicting hallucinations, which are particularly important in clinical applications (54; 57; 58). This is because medical practitioners adhere to stringent logical processes in diagnosis and avoid conclusions that contradict previous observations (59). Therefore, a VLM that accurately identifies the final diagnosis but fails to correctly respond to preceding observation-related questions would be deemed unreliable (60). Similar to prior work, our reward model in (2) addresses input- and fact-conflicting hallucination, but is distinguished by inclusion of the rule-based reward $R_S$ to address context-conflicting hallucination.

**Addressing misalignment in Vision-Language Models.** Reinforcement Learning from Human Feedback (RLHF) (17; 16; 61; 62) is a predominant paradigm for aligning VLM outputs with specific domain requirements or general human preferences. This method relies on preference data from human labelers to train a reward model, which is then used to fine-tune the VLM using reinforcement learning techniques such as Proximal Policy Optimization (PPO) (27). Our work is particularly related to approaches in the AI safety literature that aim to incorporate rule-based reward specifications into the alignment procedure (63; 64). For example, Sparrow (63) defines explicit rules to obtain more concrete feedback from human labelers and incorporates rule adherence into the training process. However, their method still requires fitting a general preference model using human or AI feedback, and their focus is on enhancing safety behavior. In contrast, we leverage rule-based specifications to concretize questions and enable automatic response labeling, eliminating the need for human-generated preference data. This approach not only reduces reliance on expensive specialist annotators but also improves model performance in terms of consistency in clinical reasoning. To the best of our knowledge, our work is among the first to apply RL-based fine-tuning to VLMs in the medical domain. We introduce a novel RL framework tailored to medical decision-making contexts by using an automatic reward function that explicitly incorporates adherence to clinical reasoning pathways into the alignment procedure.

## 4 EXPERIMENTS

We use our finetuning algorithm to develop Dr-LLaVA, a conversational VLM specialized in analyzing bone marrow pathology slides. In this Section, we describe our training and evaluation setup, and compare the performance of Dr-LLaVA with state-of-the-art VLMs in diagnosing blood cancer.

Table 1: Performance comparison with VLM baselines in single and multi-turn conversational settings. Metrics include Question-level Accuracy ($A_Q$), Conversation-level Accuracy ($A_C$), and Diagnostic Accuracy ($A_D$). Dr-LLaVA outperforms baselines across all settings.

| Model | Single-turn QA Results | | | Multi-turn QA Results | | | Diagnosis First | | | Improvised Interaction | | |
|---|---|---|---|---|---|---|---|---|---|---|---|---|
| | $A_Q$ | $A_D$ | $A_C$ | $A_Q$ | $A_D$ | $A_C$ | $A_Q$ | $A_D$ | $A_C$ | $A_Q$ | $A_D$ | $A_C$ |
| LLaVA-0-shot (2) | 16.5 | 12.6 | 0.0 | 15.2 | 11.0 | 0.0 | 16.7 | 12.6 | 0.0 | 14.7 | 12.3 | 0.0 |
| OpenFlamingo-SFT (38) | 60.5 | 55.8 | 31.3 | 81.4 | 69.9 | 46.4 | 65.2 | 55.2 | 40.3 | 70.0 | 72.0 | 41.2 |
| LLaMA-Adapter-SFT (67) | 68.6 | 70.2 | 37.8 | 70.4 | 75.4 | 42.5 | 65.2 | 74.6 | 40.2 | 66.4 | 70.0 | 43.5 |
| MiniGPT-4-SFT (39) | 64.1 | 50.0 | 32.9 | 75.8 | 75.4 | 44.2 | 66.2 | 50.0 | 40.8 | 72.2 | 71.4 | 41.6 |
| LLaVA-Med-SFT (9) | 78.2 | 76.5 | 55.6 | 91.2 | 90.3 | 85.6 | 86.2 | 82.2 | 70.8 | 85.4 | 81.3 | 71.6 |
| LLaVA-SFT (2) | 77.4 | 77.3 | 47.6 | 92.4 | 91.8 | 90.1 | 83.1 | 76.9 | 67.5 | 82.0 | 76.9 | 74.6 |
| Dr-LLaVA | **89.6** | **84.7** | **70.0** | **93.6** | **92.0** | **90.8** | **88.9** | **85.9** | **84.4** | **92.0** | **89.0** | **87.4** |

## 4.1 EXPERIMENTAL SET-UP

**Training details.** As our study concentrates on the performance of the finetuning algorithm, we base Dr-LLaVA on the same model architecture as LLaVA (2). Our LLM utilizes Vicuna-V1.5-7b (5; 6; 33), paired with the pre-trained CLIP visual encoder ViT-L/14 at an image resolution of $256 \times 256$ (65). Grid features are employed both before and after the final transformer layer to enhance the model's integration of visual data. We use a linear layer to map image features into the word embedding space, drawing on the pre-trained linear projection matrix checkpoints from LLaVA. We then conducted supervised fine-tuning for four epochs.

During the RL phase, following (66) and (18), we initialize the value model based on the LLavA-13B-based reward model. We use LoRA-based finetuning with a rank of 64 for both the attention and feed-forward network modules. Consistent with (66), we use a batch size of 512 and normalize the advantage across the batch for each PPO step. The peak learning rate was set at $3 \times 10^{-5}$, applying cosine decay, and gradients were clipped by their Euclidean norm with a threshold of 1. Training was conducted through four complete rounds using our held-out RL data. For generalized advantage estimation, we set both $\lambda$ and $\gamma$ to 1, and adopted a constant KL regularizer coefficient of 0.1. The Dr-LLaVA model was trained using four A100 80 GB GPUs.

We leverage 80% of our synthesized clinical multi-turn conversation dataset for supervised finetuning and RL and use the remaining 20% for evaluation. We split the data at the conversation level such that all question-answer pairs pertaining to a particular image belong to the same sample.

**Baselines.** We evaluate Dr-LLaVA against state-of-the-art VLMs including the LLaVA (2), Open-Flamingo (38), MiniGPT-4 (39), LLaMA-Adapter (67) and LLaVA-med (9). Given the poor zero-shot performance of these models in this specialized domain, we perform supervised finetuning for all models on our synthesized conversational data over four epochs before evaluation on the test set.

**Evaluation metrics.** We evaluate model performance using three metrics: Question-level Accuracy ($A_Q$), Conversation-level Accuracy ($A_C$), and Diagnostic Accuracy ($A_D$). $A_Q$ measures the proportion of correctly answered questions across all conversations, while $A_C$ represents the *fraction of conversations* where *all questions* were answered correctly. $A_D$ assesses the model's ability to make a correct final diagnosis, independent of its performance in preceding analysis steps.

## 4.2 RESULTS

**Dr-LLaVA outperforms VLM baselines in single and multi-turn conversations.** We start by evaluating Dr-LLaVA with single-question scenarios, focusing on instances where a clinician seeks clarification on a specific step in the image analysis process, without the model having access to prior conversational context. The results, detailed in Table 1, reveal that our finetuning algorithm significantly boosts Dr-LLaVA's performance across all metrics, outperforming state-of-the-art VLMs. Specifically, Dr-LLaVA achieved a Question-level Accuracy of 89.6%, 11.4 percentage points higher than the top baseline model, LLaVA-Med-SFT. Furthermore, Dr-LLaVA exhibited a 14.4 percentage point increase in Conversation-level Accuracy over the best baseline, even in the absence of conversational context.

Table 2: Performance under misleading clinician prompts

| Model | Metric | CQ-R | CQ-W | RQ-R | RQ-W |
|-------|--------|------|------|------|------|
| LLaVA-SFT | AQ | **99.2** | 13.8 | 91.5 | 33.3 |
|           | AD | **99.3** | 13.6 | **99.8** | 31.3 |
| Dr-LLaVA | AQ | 99.0 | **22.7** | **93.0** | **39.0** |
|          | AD | 97.9 | **33.7** | 98.6 | **48.6** |

Table 3: Impact of different reward model components

| Scenarios | Single-turn VQA | | Multi-turn VQA | |
|-----------|-----------------|------|----------------|------|
|  | $A_Q$ | $H_{cc}$ | $A_Q$ | $H_{cc}$ |
| Dr-LLaVA | **89.6** | 22.5 | **93.6** | 5.4 |
| Dr-LLaVA w/o $R_c$ | 32.4 | **1.5** | 52.1 | **0.0** |
| Dr-LLaVA w/o $R_S$ | 78.4 | 47.5 | 83.0 | 20.2 |
| Dr-LLaVA w/o $R_m$ | 85.2 | 30.6 | 87.6 | 8.8 |
| Dr-LLaVA w/o $R_l$ | 87.9 | 25.8 | 89.1 | 7.0 |
| Dr-LLaVA w/o $R_m/R_l$ | 84.2 | 33.1 | 87.2 | 10.1 |

In multi-turn conversational settings, Dr-LLaVA consistently achieves the highest performance among all models and across all metrics, demonstrating the robustness and reliability of our approach. These improvements underscore the effectiveness of our fine-tuning algorithm in ensuring that answers are consistent with clinical reasoning. Furthermore, we observe that fine-tuning generally yields substantial performance improvements within this specialized domain. This is evidenced by the markedly enhanced results of baseline models after undergoing supervised fine-tuning, in contrast to the zero-shot application of the LLaVA model, which achieves below 20% across all metrics.

These results are particularly noteworthy as they indicate that the benefits of our alignment procedure extend beyond enhancing performance in the specific conversational settings used in training. The clinical grounding provided by our framework equips the model with a more comprehensive understanding of the task, enabling superior performance even in the absence of conversational context.

**Dr-LLaVA can handle diverse styles of interactions with clinicians.** We further evaluate Dr-LLaVA in a conversational context. To capture the diverse forms of possible interactions between clinicians and VLMs, we assess all VLMs using 3 conversational scenarios: **(1) Standard Interaction (SI)** adheres to the logical dialogue sequence in Fig. 3, starting with image quality assessment and advancing through morphological analysis to reach a final diagnosis; **(2) Diagnosis First (DF)** inverts the sequence in Fig. 3, where the clinician starts by asking about the patient's diagnosis and then interacts with the model to understand the reasoning behind it; **(3) Improvised Interaction (II)** mimics the unpredictability of real-world interactions by randomizing the question sequence, presenting questions in a non-linear and potentially repetitive sequence. This is implemented by randomly sampling questions pertaining to a specific conversation with replacement.

Table 1 presents the comparative results. While LLaVA-SFT, LLaVA-Med-SFT, and Dr-LLaVA all perform well in SI conversations, Dr-LLaVA significantly outperforms the baseline models in non-traditional sequences, with performance gains ranging from 2.7 to 15.8 percentage points. Notably, Dr-LLaVA achieves conversation-level accuracy that is 13.6 percentage points higher than the best baseline in the Diagnosis First setting and 15.8 percentage points higher in the Improvised Interaction setting. This superior performance underlines Dr-LLaVA's advanced adaptive reasoning capabilities, allowing it to extract critical information from conversational contexts effectively, regardless of the question sequencing. The ability of our model to handle these varied conversational dynamics shows its potential in realistic clinical settings where dialogues may not follow a predefined order.

**Dr-LLaVA is better at correcting wrong hypotheses by clinicians.** We also evaluate the performance of the model in scenarios where physicians incorporate hypotheses into their prompts. Specifically, we examine two types of queries: Confirmation Queries (CQ), in which clinicians seek model validation for their (potentially incorrect) hypotheses, and Rationalization Queries (RQ), where clinicians provide (potentially flawed) explanations for their hypotheses and request guidance on subsequent diagnostic steps. The details for each query type are detailed in Appendix C.2.

Table 2 presents the accuracy of various VLMs in distinguishing between accurate and misleading information, with "R" indicating clinician prompts containing correct information and "W" denoting misleading prompts. All models demonstrate robust performance when clinician prompts include accurate hypotheses. However, accuracy significantly declines across all models when prompts contain misleading information. In these challenging scenarios, Dr-LLaVA consistently exhibits a higher rate of disagreement with the misleading content compared to baseline models. This suggests that our alignment algorithm enables the model to ground its responses more effectively in visual evi-

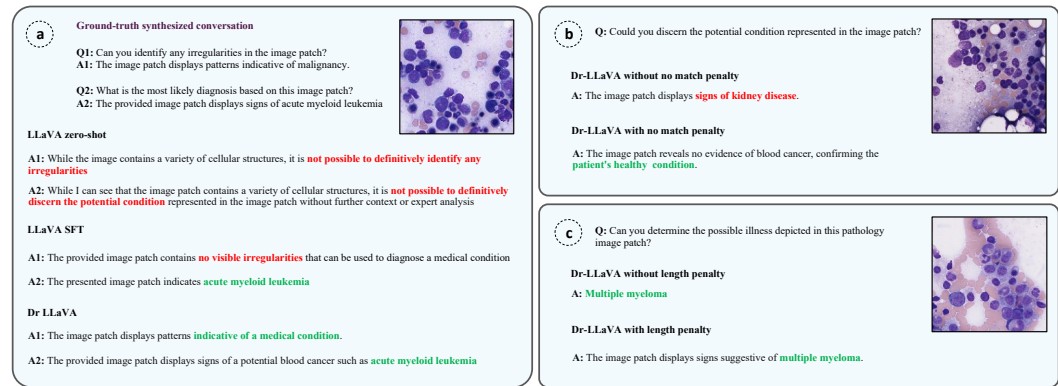

Figure 4: Qualitative examples of diagnostic responses from the Dr-LLaVA model and baselines.

dence and clinical decision-making pathways, thereby enhancing its robustness against misleading clinician queries.

**Ablation study for different components of the reward model.** We examine the impact of excluding various components of the reward model in (2). Our findings, shown in Table 3, indicate that omitting either the correctness or consistency rewards significantly reduces predictive accuracy. As expected, removing the correctness reward ($R_c$) improves answer consistency. This occurs because the model is then primarily driven to align with rule-based reasoning, disregarding the actual correctness of the responses in the context of the visual input. Eliminating the length penalty ($R_l$) and no-match penalty ($R_m$) results in moderate, yet noticeable declines in both accuracy and consistency. Qualitatively, the absence of these penalties demonstrates their vital role in preventing reward hacking and maintaining the integrity of medical dialogue. For instance, the removal of the no-match penalty causes a marked deterioration in content relevance and accuracy, with the model occasionally generating blatantly unrelated medical suggestions. An example of this is the inappropriate reference to renal conditions when analyzing bone marrow images (Fig. 4(b)). Additionally, without the length penalty, the model tends towards producing brief, often incomplete responses as observed in Fig. 4(c).

**Balancing the correctness and consistency trade-off.** The hyperparameter $\lambda$ in (2) balances the model's correctness in responding to individual questions with the overall alignment of these responses to a valid reasoning process throughout a conversation. Setting $\lambda$ to a large value imposes strong regularization on the conversational output, potentially encouraging the model to adhere to valid reasoning processes that are not grounded in the input image. For example, the model might consistently follow a (Low image quality → Inconclusive) judgment regardless of the input image. Conversely, setting $\lambda = 0$ reverts to the standard supervised fine-tuning setup, where the model optimizes solely for question-level accuracy but is likely to exhibit context-conflicting hallucinations within conversations.

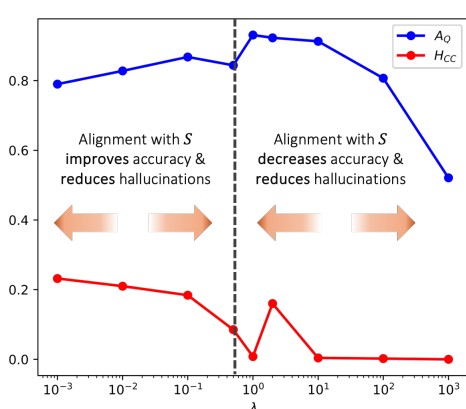

Figure 5: Impact of the hyperparameter $\lambda$.

Fig. 5 demonstrates the impact of the choice of $\lambda$ on the model performance in terms of $A_Q$ and the corresponding rate of context-conflicting hallucinations $H_{cc}$. Here, we define $H_{cc}$ as the fraction of conversations that map to invalid decision rules, i.e., $H_{cc} = E[\mathbf{1}\{\hat{s} \notin \mathcal{S}\}]$. The plot shows that increasing $\lambda$ initially improves accuracy and consistency (quantified through $H_{cc}$) reaching an optimal point beyond which further increases in $\lambda$ lead to diminished accuracy. These findings show that our alignment with valid clinical reasoning not only improves the model's coherence and trustworthiness, but can also improve the model accuracy on individual questions by regularizing the entire conversational output using prior knowledge on diagnostic scenarios.

## 5 CONCLUSION

Vision-language models (VLMs) hold the potential to become valuable tools for clinicians, researchers, and trainees, offering an interactive natural language interface for medical image analysis within clinical workflows. Yet, their utility is often compromised by the generation of "hallucinated" outputs that deviate from sound medical reasoning, leading to a lack of trust in their responses. In this paper, we introduce a novel alignment algorithm that finetunes VLMs by grounding them in rule-based representations of medical image analysis processes. This approach ensures the production of clinically valid and consistent responses across both single-turn and multi-turn interactions. Utilizing this algorithm, we develop Dr-LLaVA, a VLM specifically tailored for analyzing bone marrow image patches. Our findings reveal that Dr-LLaVA not only excels in single-turn question-answer scenarios but also demonstrates superior adaptability and accuracy in complex, multi-turn clinical dialogues, outperforming other state-of-the-art VLMs. These results suggest that grounding VLMs in structured medical analysis pathways enhances their overall clinical reasoning capabilities, making them more robust to variations in question sequencing and resilient against misleading physician hypotheses. Furthermore, this alignment strategy can be readily applied to various domains where decision-making processes can be decomposed into logical sequences of substeps, such as numerous medical tasks governed by clinical practice guidelines that codify diagnostic workflows.

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

# A   DATA

## A.1   ADDITIONAL MEDICAL CONTEXT

In this paper, we focus on the analysis of bone marrow pathology slides for the diagnosis of blood cancer disorders. Specifically, our dataset contains 512x512 pixel images of 16,340 pathology patches corresponding to healthy, inconclusive, acute myeloid leukemia, and multiple myeloma cases. The process for the analysis of bone marrow pathology slides, involves multiple steps. A pathologists has to first identify image regions that are deemed adequate for evaluation, excluding cases with either too low image quality or where the presence of other cells (e.g. red blood cells) prevents accurate medical diagnosis. Subsequently, the remaining adequate regions are examined to determine if they exhibit characteristics indicative of cancerous tissue. Specifically, in bone marrow aspirates, the assessment focuses on whether there is abnormal proliferation of cells in the regions of interest. Depending on the type of cells proliferating, the patient may be diagnosed with a corresponding hematological disorder. For instance, in our dataset, the uncontrolled proliferation of blast cells is indicative of acute myeloid leukemia, while similar proliferation of plasma cells suggests multiple myeloma.

## A.2   GENERATING A MULTI-TURN CONVERSATION DATASET

The below section provides further details on the steps we took to derive a multimodal multi-turn conversation dataset in this specialist problem domain.

**Image data**: We obtained whole pathology slide images sourced from the clinical archives of an academic medical center. These were then segmented into 512x512 pixel patches[1] and labelled as either "adequate for analysis", "particle-enriched contamination" or "blood contamination" after pathologist review. Subsequently we leveraged specialist software in order to obtain cell-counts, based on which a pathologist labelled cases with a high increase in blast or plasma cells as acute myeloid leukemia or multiple myeloma, respectively. Table A.4 details the distribution of the final diagnosis corresponding to the image data.

**Question Answer generation:** Next, we utilize the rule-based representation of the bone marrow pathology slide analysis process to create clinically meaningful multi-turn conversations. This is accomplished by filling in question and answer templates based on the respective label for each analysis step. To prevent our model from overfitting to specific expressions in these templates, we increased the diversity of questions and answers by obtaining multiple question templates from our clinical collaborators and using GPT-4 to paraphrase these templates. The respective prompts are provided below:

**Prompt for question paraphrasing:** "Perform $X$ times augmentation of the following sentence, it is for medical questions so make sure you preserve the meaning concisely."

**Prompt for answer paraphrasing** : "Perform $X$ times augmentation of the following sentence, it is for medical diagnosis so make sure you preserve the meaning concisely: 'sentence'. Also note that the question is 'question', also don't repeat anything related to in response to the question, just make sure the single sentence is grammatically correct and makes sense."

Table A.4: Distribution of Final Diagnoses in the Pathology Slide Image Dataset

| Diagnosis | Number |
|---|---|
| Blood contamination | 10083 |
| Particle enriched contamination | 3510 |
| Acute myeloid leukemia | 1531 |
| Multiple myeloma | 932 |
| Healthy | 284 |

---

[1]During training, we resize the image to a resolution of 256x256 pixels before feeding it into the image encoder.

# B INSTRUCTION TUNING DETAILS

## B.1 VLM RESPONSE LABELLING

In this work we leverage a simple rule-based reward model that evaluates the correctness of LLM responses based on the presence of relevant keywords in their answer. The respective keywords are depicted in Table B.5. For a certain keyword to be valid we require it to appear without negation. An answer is classified as 'no match' in case it does not contain any of the considered keywords for the respective analysis step.

## B.2 TRAINING DETAILS

As our study concentrates on the performance of the finetuning algorithm, we base Dr-LLaVA on the same model architecture as LLaVA (2). Our LLM utilizes Vicuna-V1.5-7b (5; 6; 33), paired with the pre-trained CLIP visual encoder ViT-L/14 at an image resolution of $256 \times 256$ (65). Grid features are employed both before and after the final transformer layer to enhance the model's integration of visual data. We use a linear layer to map image features into the word embedding space, drawing on the pre-trained linear projection matrix checkpoints from LLaVA. We then conducted supervised fine-tuning for four epochs.

During the RL phase, following (66) and (18), we initialized the value model based on the LLavA-13B-based reward model. We used LoRA-based finetuning with a rank of 64 for both the attention and feed-forward network modules. Consistent with (66), we used a batch size of 512 and normalized the advantage across the batch for each PPO step. The peak learning rate was set at $3 \times 10^{-5}$, applying cosine decay, and gradients were clipped by their Euclidean norm with a threshold of 1. Training was conducted through four complete rounds using our held-out RL data. For generalized advantage estimation, we set both $\lambda$ and $\gamma$ to 1, and adopted a constant KL regularizer coefficient of 0.1. The Dr-LLaVA model was trained using four A100 80 GB GPUs.

We leverage 80% of our synthesized clinical multi-turn conversation dataset for supervised finetuning and RL and use the remaining 20% for evaluation. We split the data at the conversation level such that all question-answer pairs pertaining to a particular image belong to the same sample. We use different prompt templates and rephrasing for the question-answer pairs in the training and testing sets to ensure that the models do not over-fit to specific phrasing of the clinician-VLM conversations

Table B.5: Keywords considered in rule-based reward model

| Analysis Steps | Classification | Keywords |
|---|---|---|
| Image Quality Assessment | High quality | effective, appropriate, suitable, sufficient, optimal |
| | Low quality | not, no, inadequate, unsuitable |
| | No Match | - |
| Cell Quality Assessment | Adequate | optimal, advantageous, suitable, adequate, well, prime |
| | Blood | blood, RBC |
| | Clot | particles |
| | No Match | - |
| Cell Abnormality Analysis | Normal | normal, healthy, no abnormality |
| | Abnormal | cancer, disorder, malignancy |
| | Inadequate | low, subpar, inadequate |
| | No Match | - |
| Detailed Cell Proliferation Reasoning | Blast Cell Proliferation | myeloblast |
| | Plasma Cell Proliferation | plasma cells |
| | Normal | no abnormal, no proliferation, normal |
| | Inadequate | low, subpar, inadequate |
| | No Match | - |
| Final Diagnosis | Healthy | no malignancy phenotype, healthy |
| | Acute Myeloid Leukemia | acute myeloid leukemia, AML |
| | Multiple Myeloma | multiple myeloma, MM |
| | Inconclusive | low quality, inadequate |
| | No Match | - |

# C EVALUATION

## C.1 EVALUATION METRICS

To effectively assess the performance of our proposed model, we measure the accuracy of our model at the question, conversation and diagnosis level.

1. **Question-level Accuracy** ($A_Q$): This metric evaluates the model's performance at the single question level. It is obtained by dividing the number of questions answered correctly by the total number of questions:

$$A_Q = \frac{\text{Number of correct answers}}{\text{Total number of questions}} \tag{4}$$

2. **Conversation-level Accuracy** ($A_C$): This metric assesses the model's accuracy at the conversation level. Here we only consider a VLM's response as correct if it is able to correctly answer all questions pertaining to a multi-turn conversation about a specific case.

$$A_C = \frac{\text{Number of conversations with all questions answered correctly}}{\text{Total number of cases}} \tag{5}$$

This metric asses the model's capability to consistently provide accurate answers across all questions within a multi-turn conversation, enabling the model to be a trustworthy companion throughout the full image analysis process.

3. **Diagnostic Accuracy** ($A_D$): This metric focuses solely on the VLMs' responses to questions about the final diagnosis, as this is often the primary concern for medical decision-makers:

$$A_D = \frac{\text{Number of correctly answered diagnosis questions}}{\text{Total number of cases}} \tag{6}$$

In conclusion, these three distinct levels of accuracy—$A_Q$, $A_C$, and $A_D$—provide a comprehensive evaluation of the proposed model's effectiveness in handling different aspects of medical inquiries. By breaking down the analysis to question, conversation, and diagnosis levels, we can better understand the model's strengths and pinpoint areas for improvement in handling complex medical scenarios.

## C.2 EXAMPLE PROMPTS TO EVALUATE MODEL PERFORMANCE GIVEN CLINICIAN HYPOTHESES

This section presents example prompts crafted to evaluate our model's ability to respond to scenarios where physicians incorporate hypotheses into their prompts. The prompts are divided into two categories: Confirmation Queries (CQ) and Rationalization Queries (RQ).

CONFIRMATION QUERY (CQ) PROMPTS

Confirmation Queries aim to assess the model's ability to validate clinician opinions. These queries challenge the model to either concur with or contest a clinician's judgment, which may be accurate (CQ-R) or erroneous (CQ-W).

**Example Prompt 1 (CQ-R):** "After reviewing the image, the clinician believes that [correct statement]. Do you agree with this assessment?"

**Example Prompt 2 (CQ-W):** "After examining the image, the clinician suggests that [misleading statement]. Do you concur with this opinion?"

RATIONALIZATION QUERY (RQ) PROMPTS

Rationalization Queries present the model with a previous conclusion, which may be correct (RQ-R) or incorrect (RQ-W), and ask about the next diagnostic steps. These queries assess the model's ability to correct incorrect hypotheses even when not explicitly prompted to do so.

**Example Prompt 3 (RQ-R):** "A previous clinician reviewed the image and concluded that [accurate rationale]. Considering this, what would be your next step in the diagnostic process? [Question]"

**Example Prompt 4 (RQ-W):** "A previous clinician interpreted the image and believed [erroneous rationale]. With this in mind, how would you proceed with the diagnosis? [Question]"

### C.3 ASSESSING HALLUCINATION RATES

In this work we focus on context-conflicting hallucinations, which we define as an answer that deviates from the expected pathways outlined in the rule-based representation of the pathology slide analysis process, as illustrated in 3. We use a rule-based reward model to classify the VLM responses according to the possible choices within the rule-based representation of medical reasoning. This allows us to quantify the proportion of answers that do not follow any logical trajectory within this framework.

