# OpenReview forum: "Aligning Multimodal Models for Clinical Reasoning using Rule-based Rewards"
_ICLR.cc/2025/Conference — ICLR 2025 Conference Withdrawn Submission_

### Official Review · Reviewer_ABWK · 2024-10-29

**Soundness:** 3
**Presentation:** 1
**Contribution:** 2
**Rating:** 3
**Confidence:** 5

**Summary:**

This paper used rule-based rewards generated from clinical diagnostic pathways instead of human feedback based rewards for reinforcement learning to train a medical VLM, Dr-LLaVA. The rule-based rewards include consistency and correctness. 16,340 bone marrow pathology patches and hematopathologist's annotations were used to generate conversations for training. Dr-LLaVA performed better across all settings when compared to baseline models which are mostly further supervised fine-tuned with the same conversations but without the reward function.

**Strengths:**

1. Rather than RLHF, the paper uses a hierarchy set of rules based on clinical reasoning pathways to train a medical VLM. This approach is a way around the lack of human annotations.
2. Not only conventional conversational settings, but also diagnosis related and random order question sequence settings were tested and Dr-LLaVA performs better than the baseline models on all settings.

**Weaknesses:**

1. GPT-4 is used to generate the conversations, and a rigorous evaluation is done to ensure no hallucinations are introduced. However, the authors provide no details on the specific evaluation methods used to ensure this, which limits the reproducibility of the work. Additionally, this lack of transparency undermines the objective of reducing hallucinations with reinforcement learning, as the GPT-4-generated dataset could still contain incorrect statements.
2. The essential details on clinical decision making pathways, dataset generation, and hallucination evaluations are either missing or located in the appendix rather than the main text, making it challenging to fully understand the methodology without the appendix. Since these are important in the work, the absence from the main text disrupts the flow and limits the study’s context and reproducibility. Including these details in the main body, or at least providing a summary with clear references to the appendix for further depth, would make the paper more accessible and cohesive.
3. The presentation of the work needs improvement, especially in organization and clarity to be publishable quality. Currently, the layout requires readers to frequently go back and forth, as text descriptions are located far from the tables and figures that they describe. This disrupts the flow and makes it challenging to follow the paper. For instance, Figure 3 is introduced in Section 2 before Figure 2, which creates confusion in the narrative structure. Additionally, abbreviations are often used without prior definition; for example, Table 2 and Table 3 feature abbreviated column headings like "CQ-R" and "Hcc," but their meanings are not clarified until much later in the text or, in some cases, are only defined in the Appendix. Including these definitions upfront or alongside the tables would enhance readability. Also, typos, such as "ruke" instead of "rule" on line 144 needs to be corrected.
4. The paper lacks experiments specifically designed to address the issue of misalignment between image and text and to demonstrate how Dr-LLaVA mitigates this problem, despite this being a primary motivation for using RL over SFT. However, without empirical evidence or ablation study showing that RL successfully reduces misalignment, the paper's contribution is weakened.
5. The paper lacks a comprehensive qualitative evaluation, presenting only a single qualitative example set in Figure 4. This limited approach does not provide enough insight into the model's practical application and performance across varied cases. Conducting a broader qualitative evaluation, ideally involving human assessments by clinicians, would allow for a more in-depth assessment of the model's reliability and relevance in real-world clinical settings.
6. The paper lacks a section on limitations, which is essential for providing a balanced view of the work's contributions and areas for future improvement.
7. The paper is missing an ethics statement, which is essential in such a research involving patient information. An ethics statement provides transparency about how ethical considerations were addressed, including data handling, privacy safeguards, and compliance with relevant regulations.

**Questions:**

1. How many whole slide images did you use to generate 16K datasets?
2. Do you plan to release the model and the dataset?
3. What is the source of the clinical reasoning pathways?

**Details Of Ethics Concerns:**

The paper is missing an ethics statement, which is essential in such a research involving patient information which is sensitive dataset.

---

### Official Review · Reviewer_EMu9 · 2024-11-01

**Soundness:** 3
**Presentation:** 3
**Contribution:** 3
**Rating:** 5
**Confidence:** 4

**Summary:**

Vision-language models often generate textual outputs that are not grounded in the input image-text information. The authors propose a new alignment algorithm that uses rule-based representations of clinical reasoning (obtained from a physician) to ground VLMs in medical knowledge. These representations are used to (i) generate visual instruction tuning data and (ii) derive a rule-based reward function that evaluates clinical validity of VLM responses. The resulting algorithm eliminates the need for human involvement in data construction or reward model construction. The algorithm is used to develop Dr-LLaVA, a conversational VLM fine-tuned for analyzing bone marrow pathology slides. Dr-LLaVA is shown to outperform several VLM baselines and can handle diverse styles of interactions with clinicians.

**Strengths:**

1. The paper introduces an approach for improving the quality of textual outputs generated by vision-language models in the medical setting. The proposed approach intends to mimic clinical reasoning rules typically employed by physicians.
2. The proposed approach is capable of understanding/generating multi-turn conversations in the medical setting, unlike many existing medical VLMs. The proposed approach can also handle diverse styles of interactions with clinicians.
3. The authors demonstrate that their approach leads to performance improvements over several existing methods in this domain and conforms closely with clinical reasoning pathways.
4. The paper is clearly written and is well organized.

**Weaknesses:**

1. **Limited Scalability:** The application area explored in this paper is narrow in scope (bone marrow pathology classification), and it seems that in order to expand to other medical applications, a lot of manual effort is necessary. In particular, (1) a physician will have to manually generate a set clinical reasoning rules in order to extend this approach to other medical application areas and (2) the rule-based reward model will need a new set of manually-constructed keywords (as in Table B.5). This weakness limits the widespread usability of this approach.

2. **Additional Methodology Details:** Some important details with respect to the construction of the dataset and the rule-based rewards are missing, as detailed below:

    a. **Multi-Turn Conversations:** Each conversation in the synthesized conversational dataset consists of five question-answer pairs (Line 135). Why do all conversations include exactly five QA pairs, when it’s possible that the conversation could end in fewer turns if a leaf node of the decision tree is reached early (e.g. low quality image)?

    b. **Dataset details:** No details with respect to the composition of the synthesized dataset are provided. How large is the dataset? What is the class distribution with respect to the diagnoses? Are there sufficient samples for each of the possible decision pathways in the clinical reasoning decision tree?

    c. **Keyword Matching:** The authors use a manually pre-defined list of keywords in order to compute rewards and accuracy metrics. How accurate is the process of keyword matching, and how often do false positives / negatives arise?

3. **Need for finer-grained evaluations:** The performance metrics in Table 1 are aggregate metrics. How well does the proposed approach perform in comparison to baselines across each diagnosis and each possible decision pathway?

**Questions:**

In addition to the weaknesses detailed above, my additional questions are listed below:

1. In Line 136, the authors state that they conduct “a rigorous evaluation to ensure that no hallucinations are introduced within the template set.” Can the authors provide additional details with respect to this evaluation and the numbers/types of hallucinations identified?

2. To the best of my understanding, the single-turn QA performance values reported in Table 1 for question-level accuracy ($A_Q$) and conversation-level accuracy ($A_C$) should be equal, since there is only one question per conversation. Why are the values nearly 20 points apart for Dr-LLaVA?

---

### Official Review · Reviewer_SSVZ · 2024-11-04

**Soundness:** 3
**Presentation:** 3
**Contribution:** 3
**Rating:** 8
**Confidence:** 2

**Summary:**

This paper presents a new alignment algorithm for VLMs in clinical settings, addressing hallucination where models generate inaccurate or inconsistent responses. The authors propose a rule-based approach that grounds VLMs in clinical reasoning by creating large-scale visual instruction data and developing a reward function to ensure responses align with medical knowledge across multi-turn conversations. This avoids high costs of RLHF and they used it to create Dr-LLaVA.

**Strengths:**

Strong and sound method with clear performance improvements over many experiments.

**Weaknesses:**

Small things:
- Figure 1 needs to be bigger. The conversation text cannot be read unless I zoom in alot.
- Typo "rule": line 144 our conversational diagnostic system leverages ruke-based representations (Fig. 3) - should be "rule-based"

**Questions:**

What does line 136: "followed by a rigorous evaluation to ensure no hallucinations are introduced within the template set" mean specifically? What was the criteria used in this evaluation, or what were some examples hallucinations / examples of how hallucinations were checked in the templates.

---

### Official Review · Reviewer_6HNL · 2024-11-04

**Soundness:** 3
**Presentation:** 3
**Contribution:** 2
**Rating:** 3
**Confidence:** 4

**Summary:**

* **Topic**: Medical VLMs, Rule-based Rewards, RLHF
* **Summary**: This paper proposed to improve medical VLMs through a rule-based framework. It includes (i) using a rule to generate instruction tuning data, (ii) using the rule to evaluate the response, and (iii) using the rule-based rewards to perform RLHF to reduce hallucination. The experimental results show that the proposed approach achieves better performance than OpenFlamingo, LLaVA, and LLaVA-Med, etc.

**Strengths:**

* **Methodology**: In general, it's a sound approach. By designing the heuristics, the approach enables a rule-based system to **produce** training datasets, **evaluate** the responses, and **post-train** the models using RLHF. Similar insights were also approved in [1].

* **Writing**: The paper is well-written and easy to follow. The experimental designs can support the claims in this paper.

[1] Mu, Tong, et al. "Rule-Based Rewards for Language Model Safety." Advances in Neural Information Processing Systems (2024).

**Weaknesses:**

* One of my biggest concerns about this paper is that it didn't show the potential of such an approach in medical **generative** VLMs. To be specific, the modeling can be replaced with a **discriminative** classification model since the responses from the MVLM are "pseudo" free-text responses, and they can be replaced with text templates + classification labels. So is a **large** generative VLM (7B parameters) really needed to solve the proposed task?

**Questions:**

* Is it possible to have small discriminative image-text models as the baselines (e.g., CLIP [1] and ALBEF [2])?

[1] Radford, Alec, et al. "Learning transferable visual models from natural language supervision." International conference on machine learning. PMLR, 2021.

[2] Li, Junnan, et al. "Align before fuse: Vision and language representation learning with momentum distillation." Advances in neural information processing systems 34 (2021): 9694-9705.

---

### Note · Authors · 2024-11-23

I have read and agree with the venue's withdrawal policy on behalf of myself and my co-authors.